# Global mapping of highly pathogenic avian influenza H5N1 and H5Nx clade 2.3.4.4 viruses with spatial cross-validation

Madhur S Dhingra[1,2†], Jean Artois[1†], Timothy P Robinson[3], Catherine Linard[1,4], Celia Chaiban[1], Ioannis Xenarios[5,6], Robin Engler[5], Robin Liechti[5], Dmitri Kuznetsov[5], Xiangming Xiao[7,8,9], Sophie Von Dobschuetz[10], Filip Claes[11], Scott H Newman[12]*, Gwenaëlle Dauphin[10]*, Marius Gilbert[1,13]*

[1]Spatial Epidemiology Lab, Université Libre de Bruxelles, Brussels, Belgium; [2]Department of Animal Husbandry and Dairying, Government of Haryana, Panchkula, India; [3]Livestock Systems and Environment, International Livestock Research Institute, Nairobi, Kenya; [4]Department of Geography, Université de Namur, Namur, Belgium; [5]Swiss-Prot and Vital-IT group, Swiss Institute of Bioinformatics, Lausanne, Switzerland; [6]Center for Integrative Genomics, University of Lausanne, Lausanne, Switzerland; [7]Department of Microbiology and Plant Biology, University of Oklahoma, Norman, United States; [8]Center for Spatial Analysis, University of Oklahoma, Norman, United States; [9]Institute of Biodiversity Science, Fudan University, Shanghai, China; [10]Animal Production and Health Division, Food and Agriculture Organization of the United Nations, Rome, Italy; [11]Emergency Center for Transboundary Animal Diseases, FAO Regional Office for Asia and the Pacific, Bangkok, Thailand; [12]Emergency Center for Transboundary Animal Diseases, Food and Agriculture Organization of the United Nations, Hanoi, Vietnam; [13]Fonds National de la Recherche Scientifique, Brussels, Belgium

*For correspondence: scott. newman@fao.org (SHN); Gwenaelle.Dauphin@fao.org (GD); marius.gilbert@gmail.com (MG)

[†]These authors contributed equally to this work

Competing interests: The authors declare that no competing interests exist.

**Abstract** Global disease suitability models are essential tools to inform surveillance systems and enable early detection. We present the first global suitability model of highly pathogenic avian influenza (HPAI) H5N1 and demonstrate that reliable predictions can be obtained at global scale. Best predictions are obtained using spatial predictor variables describing host distributions, rather than land use or eco-climatic spatial predictor variables, with a strong association with domestic duck and extensively raised chicken densities. Our results also support a more systematic use of spatial cross-validation in large-scale disease suitability modelling compared to standard random cross-validation that can lead to unreliable measure of extrapolation accuracy. A global suitability model of the H5 clade 2.3.4.4 viruses, a group of viruses that recently spread extensively in Asia and the US, shows in comparison a lower spatial extrapolation capacity than the HPAI H5N1 models, with a stronger association with intensively raised chicken densities and anthropogenic factors.

## Introduction

In 1996, highly pathogenic avian influenza of subtype H5N1 gave rise to the progenitor of the present H5N1 HPAI subtype in Guangdong, China (A/Goose/Guangdong/1/96[H5N1]) (**Duan et al.,**

*2007*). Initially, restricted to Southern China, the virus started spreading in 2003 and by 2008, it had spread to more than 60 countries (*Food and Agriculture Organization of the United Nations, 2016*), persisting now in only a few. Transmission of infection from birds to humans was also reported, causing disease in 850 confirmed human cases including 449 deaths, as of May 2016, making this virus a continuing source of human health concerns (*WHO/GIP, 2016*; *Lai et al., 2016*). Reassortment with other influenza viruses led to the replacement of most internal viral genes of the original H5N1 virus. However, the haemagglutinin (HA) gene H5 has remained present in all isolates and was therefore used to develop a standardised 'clade' nomenclature, first adopted in 2008, based on the evolution and divergence of H5N1 viruses that evolved from the original HA gene of the 1996 H5N1 virus (*WHO/OIE/FAO H5N1 Evolution Working Group, 2008*). In the initial years from 1996 to 2008, 10 distinct clades (0–9) had been generated and by 2012, 11 distinct actively circulating clades had been identified (*World Health Organization/World Organisation for Animal Health/Food and Agriculture Organization (WHO/OIE/FAO) H5N1 Evolution Working Group, 2014*).

Between 2009 and 2013, H5Nx HPAI viruses from the clade 2.3.4 showed an apparent geographical range expansion and were not only of the H5N1 subtype. Continuous live-poultry market surveillance in China identified novel clade 2.3.4 reassortant viruses of different H5N2, H5N5 and H5N8 subtypes, alongside H5N1 (*Gu et al., 2011*; *Zhao et al., 2012*, *2013*). All these viruses were part of an H5 monophyletic group of viruses that shared the H5 gene of an H5N1 clade 2.3.4 variant with neuraminidase (NA) genes from different viruses (*Gu et al., 2013*). Consequently, the nomenclature of H5Nx viruses that clustered in this divergent HA group was updated as a new clade 2.3.4.4, in addition to two other new clades (*Smith and Donis, 2015*). From January 2014 onward, viruses of clade 2.3.4.4 started spreading internationally. The first H5N8 HPAI virus outbreaks outside of China were reported in South Korea and Japan in spring 2014 (*Hill et al., 2015*). In May 2014, a novel H5N6 clade 2.3.4.4 reassortant caused outbreaks in China and Lao PDR (*Wong et al., 2015*) and thereafter from Viet Nam and Myanmar. In November 2014, H5N8 HPAI viruses were reported from Germany, Netherlands, UK, Italy and the Russian Federation in rapid succession. In the same autumn and winter 2014/2015, H5N2 HPAI were reported from outbreaks in British Colombia, Canada. This virus contained genes similar to those of the Eurasian clade 2.3.4.4 alongside genes from North American wild bird lineages (*World Organisation for Animal Health, 2016* ). In Taiwan, novel (H5N2, H5N3) reassortants also caused several outbreaks in 2014 (*Lee et al., 2016*). In December 2014, the new H5N8 and H5N2 HPAI viruses were detected in wild birds in Washington USA, before being found in poultry. By February 2015, the H5N2 HPAI virus had triggered a true epidemic in commercial poultry in the US, with nearly 43 million chickens and 7 million turkeys killed or culled across more than 20 different states (*Poultry Science Association, 2016*). All these HPAI H5N1, H5N2, H5N6 and H5N8 viruses found in Eurasia and North America shared an H5 gene segment belonging to clade 2.3.4.4 (*Claes et al., 2014*; *Food and Agriculture Organization of the United Nations, 2016*).

In summary, we can describe two periods and groups of viruses. From 2003 to 2010, the H5 HPAI viruses responsible for international spread and most outbreaks in poultry were of the N1 type, with continuous evolution of the H genes into different sub-lineages and gradual changes in its internal genes yielding clades and sub clades. From 2010 onward, H5Nx clade 2.3.4 viruses reassorted with several other avian influenza viruses leading to generation of a diversity of H5 clade 2.3.4.4 viruses bearing NA other than N1 in Asia. These novel reassortant viruses then began spreading internationally in 2013, in some cases further reassorting with viruses from other geographic lineages to yield new viruses, all bearing an H5 clade 2.3.4.4 haemagglutinin.

Following the spatio-temporal pattern of H5N1 HPAI spread, several spatial analytical studies were conducted to identify risk factors associated with H5N1 HPAI presence. The majority of these have been country-level studies in Thailand (*Gilbert et al., 2006*), Viet Nam (*Minh et al., 2009*), China (*Martin et al., 2011*), Bangladesh (*Ahmed et al., 2012*), Indonesia (*Yupiana et al., 2010*) and India (*Dhingra et al., 2014*). Several studies have also been conducted at regional (*Adhikari D, 2009*; *Gilbert et al., 2008*; *Williams and Peterson, 2009*) and continental levels (*Hogerwerf et al., 2010*; *Peterson and Williams, 2008*). Spatial risk factors associated with H5N1 HPAI presence through different studies were reviewed in 2012 (*Gilbert and Pfeiffer, 2012*) and the study highlighted domestic duck density, indices of water presence (distance to rivers and proportion of land occupied by water) and anthropogenic variables (human population density and distance to

roads) to be the most consistent risk factors across studies, countries and scales. However, studies comparing different sets of factors were never carried out at a global scale, and none made a distinction between clades and sub-lineages.

In this analysis, we aimed to produce a first global suitability map for H5N1 HPAI virus sustained transmission, to establish its capacity to provide reliable spatial extrapolations at large spatial scales and to compare different sets of spatial predictor variables in their predictive capacity. Machine learning techniques have become very powerful in reproducing observed distribution patterns with sets of predictor variables, but their skill in spatial extrapolation is rarely quantified and could help better discriminate among sets of important predictor variables. In addition, the very fast recent spread of clade 2.3.4.4 H5Nx viruses (H5N1, H5N2, H5N6 and H5N8), associated with multiple reassortments was unprecedented (*De Vries et al., 2015*) and warranted further examination. A separate analysis of how 2.3.4.4 H5Nx viruses had spread in the geographical and environmental space was hence carried out in comparison to the HPAI H5N1 viruses.

## Results

Boosted Regression Trees (BRT) models were developed to predict the global suitability of H5N1 HPAI and H5Nx clade 2.3.4.4 presence. The predictor variables were categorised into four sets (*Table 1*) of variables. The Set 1 variables included the host variables of extensive and intensive chicken densities, human population density, and a variable to account for the effect of mass vaccination of poultry in China (IsChina). Set 2 included land cover variables with IsChina. Set 3 included Fourier-transformed climatic variables of land-surface temperature (LST) and Normalised Difference Vegetation Index (NDVI) with IsChina. Finally, Set 4 variables included Set 1 variables in addition to selected variables from the earlier sets that were selected on the basis of prior epidemiological knowledge. The models were subjected to three different types of cross validations to measure their goodness-of-fit (GOF) and transferability: (i) standard cross-validation (CV) with a random and stratified divide between training and validation sets, (ii) a calibrated cross-validation to account for the spatial sorting bias (SSB) *sensu Hijmans (2012)* i.e. the tendency to have distance between training-presence and testing-presence sites to be smaller than the distance between training-presence and testing-absence sites, and (iii) a spatial cross-validation (Spatial CV) to spatially separate the training and validation sets by large distances and measure the spatial extrapolation capacity of the models.

The bootstrapped goodness of fit values for the H5N1 HPAI and H5Nx HPAI clade 2.3.4.4 models for the different sets of covariates and cross validation methods are shown in *Figure 1*. For the H5N1 HPAI global model, all overall GOF metrics were good with predictive accuracy Area Under the Curve (AUC) values higher than 0.9 when evaluated through standard CV (*Figure 1*). The reduction in GOF taking into account the SSB was minor and followed the same pattern. However, when evaluated through spatial CV, the different sets of covariates showed contrasting AUC values. The land-use (Set 2) and eco-climatic (Set 3) based models extrapolated poorly, and the Set 1 and Set 4 performed best. It is also noteworthy, that the combination of Sets 1 and 2 (Set 2.1), or Sets 1 and 3 (Set 3.1) did not result in significantly better models than Set 1 alone (*Figure 1—figure supplement 1*), and even tended to reduce the average AUC of spatial CV.

The models for the H5Nx clade 2.3.4.4 virus also had high GOF metrics estimated by standard CV (*Figure 1*). Here too, a significant amount of predictive power was already obtained with the models containing only Set 1 variables, with AUC values close to 0.9. There was a strong impact of spatial CV on the GOF metrics, with a drastic reduction in predictive power when extrapolating over large distances (*Figure 1*). Throughout the different spatial CV metrics, Set 2, and 4 showed better AUC values than Set 1, and given that Set 4 was more parsimonious, with fewer predictor variables, it was kept as the final model for H5Nx clade 2.3.4.4 suitability. Similar conclusions could be drawn from models using combinations of Set 1 and Set 2 (Set 2.1), or Set 1 and Set 3 (Set 3.1) (*Figure 1—figure supplement 1*).

The relative contribution (RC) of the predictor variables of Set 1 and Set 4 for H5N1 HPAI and the H5 HPAI clade 2.3.4.4 models are presented in *Figure 2*. The most noticeable difference concerned the role of domestic duck density, human population density and chicken density. The H5Nx HPAI clade 2.3.4.4 showed much higher RC for human population density and intensively raised chickens than the H5N1 HPAI one. Conversely, a comparatively much higher RC of domestic duck density and extensively raised chicken was observed for the H5N1 HPAI model than for the H5Nx HPAI

**Table 1.** List of predictor variables used for modelling the suitability of HPAI H5N1 and H5Nx clade 2.3.4.4 viruses using BRT models.

| Set | Variable full name | Abbreviation | Source |
|---|---|---|---|
| **Set 1: Host Variables** | | | |
| | Duck density | DuDnLg | *Robinson et al. (2014)* |
| | Extensive Chicken Density | ChDnLgExt | *Gilbert et al. (2015)* |
| | Intensive Chicken Density | ChDnLgInt | *Gilbert et al. (2015)* |
| | Human Population Density | HpDnLg | *Linard et al. (2012)*; *Gaughan et al. (2013)*; *Sorichetta et al. (2015)*; CIESIN's GPW Database |
| | Vaccination in China | IsChina | FAO Global Administrative Unit Layers (GAUL) database |
| **Set 2 - Land Cover Variables** | | | |
| | Evergreen Deciduous Needleleaf Trees | EDNTrees | *Tuanmu and Jetz (2014)* |
| | Evergreen Broadleaf Trees | EBTrees | *Tuanmu and Jetz (2014)* |
| | Deciduous Broadleaf Trees | DBTrees | *Tuanmu and Jetz (2014)* |
| | Mixed/Other Trees | MixedTrees | *Tuanmu and Jetz (2014)* |
| | Shrubs | Shrubs | *Tuanmu and Jetz (2014)* |
| | Herbaceous Vegetation | HerbVeg | *Tuanmu and Jetz (2014)* |
| | Cultivated and Managed Vegetation | CultVeg | *Tuanmu and Jetz (2014)* |
| | Regularly Flooded Vegetation | RegFlVeg | *Tuanmu and Jetz (2014)* |
| | Urban/Built-up | UrbanBltp | *Tuanmu and Jetz (2014)* |
| | Open Water | Owat | *Tuanmu and Jetz (2014)* |
| | Distance to Water | Dwat | - |
| | Vaccination in China | IsChina | FAO Global Administrative Unit Layers (GAUL) database |
| **Set 3- Eco-climatic Variables** | | | |
| | Day LST* Annual mean | Tmp | *Scharlemann et al. (2008)* |
| | Day LST Amplitude annual | TmpAmp1an | *Scharlemann et al. (2008)* |
| | Day LST Amplitude bi-annual | TmpAmp2an | *Scharlemann et al. (2008)* |
| | Day LST Amplitude tri-annual | TmpAmp3an | *Scharlemann et al. (2008)* |
| | Day LST Variance annual | TmpVar1an | *Scharlemann et al. (2008)* |
| | Day LST Variance bi-annual | TmpVar2an | *Scharlemann et al. (2008)* |
| | Day LST Variance annual, bi and tri-annual | TmpVar123an | *Scharlemann et al. (2008)* |
| | NDVI[†] Annual mean | NDVI | *Scharlemann et al. (2008)* |
| | NDVI Amplitude annual | NDVIAmp1an | *Scharlemann et al. (2008)* |
| | NDVI Amplitude bi-annual | NDVIAmp2an | *Scharlemann et al. (2008)* |
| | NDVI Amplitude tri-annual | NDVIAmp3an | *Scharlemann et al. (2008)* |
| | NDVI Variance annual | NDVIVar1an | *Scharlemann et al. (2008)* |
| | NDVI Variance bi-annual | NDVIVar2an | *Scharlemann et al. (2008)* |
| | NDVI Variance tri-annual | NDVIVar3an | *Scharlemann et al. (2008)* |
| | NDVI Variance annual, bi and tri-annual | NDVIVar123an | *Scharlemann et al. (2008)* |
| | Vaccination in China | IsChina | FAO Global Administrative Unit Layers (GAUL) database |
| **Set 4: Risk-based selection of variables** | | | |
| | Duck density | DuDnLg | *Robinson et al. (2014)* |
| | Extensive Chicken Density | ChDnLgExt | *Gilbert et al. (2015)* |
| | Intensive Chicken Density | ChDnLgInt | *Gilbert et al. (2015)* |

*Table 1 continued on next page*

*Table 1 continued*

| Set | Variable full name | Abbreviation | Source |
|---|---|---|---|
| | Human Population Density | HpDnLg | *Linard et al. (2012)*; *Gaughan et al. (2013)*; *Sorichetta et al. (2015)*; CIESIN's GPW Database |
| | Cultivated and Managed Vegetation | CultVeg | *Tuanmu and Jetz (2014)* |
| | Open Water | Owat | *Tuanmu and Jetz (2014)* |
| | Distance to Water | Dwat | - |
| | Day LST annual mean | Tmp | *Scharlemann et al. (2008)* |
| | Vaccination in China | IsChina | FAO Global Administrative Unit Layers (GAUL) database |

*LST = Land Surface Temperature, †NDVI = Normalised Difference Vegetation Index

clade 2.3.4.4 models. Upon the inclusion of additional predictors in Set 4 (*Figure 2*), the influence of these host-based predictor variables followed a similar pattern. In addition, annual mean temperature made a relatively high contribution in both models, and cultivated vegetation showed a much higher RC in the H5Nx HPAI clade 2.3.4.4 model than in the H5N1 HPAI one.

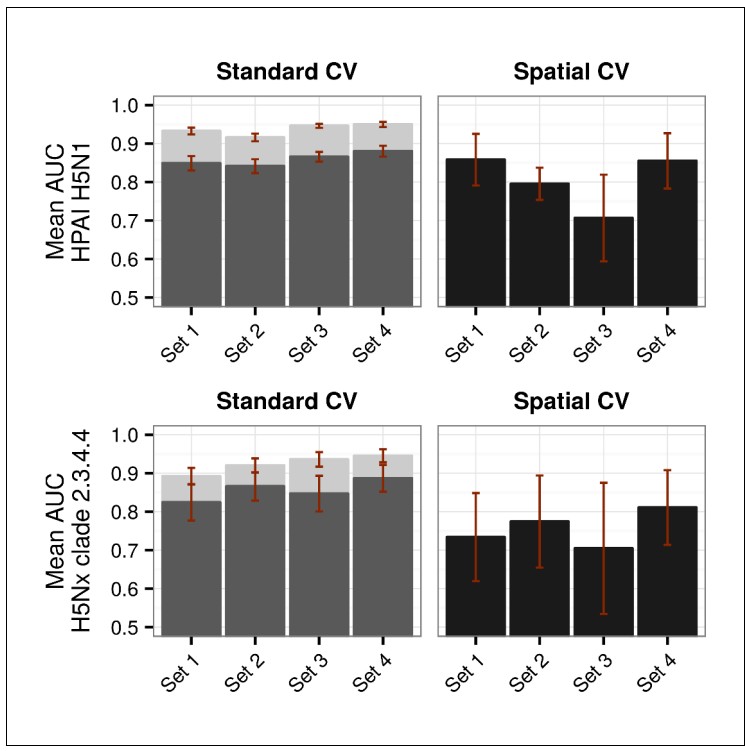

**Figure 1.** Representation of Area under Receiver Operating Curve (AUC) values for HPAI H5N1 and H5Nx models. Representation of AUC values for HPAI H5N1 and New Clade H5Nx 2.3.4.4 model for all sets of predictor variables, assessed through standard cross validation (Standard CV), in light grey, and accounting for spatial sorting bias (SSB) in dark grey. On the right, the AUC values for spatial cross validation (Spatial CV) are represented in black. All these metrics represent mean AUC ± standard deviation. Additionally, the AUC values for Set 2.1 and Set 3.1 are represented in *Figure 1—figure supplement 1*.
The following figure supplement is available for figure 1:

**Figure supplement 1.** Comparison of AUC values of additional sets (Set 2.1 and Set 3.1) of predictor variables.

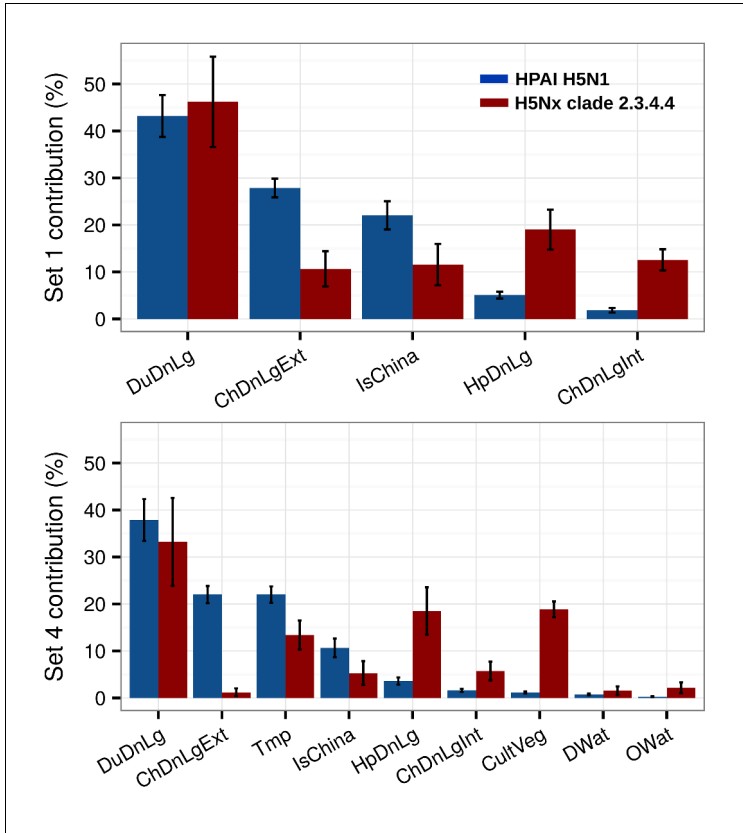

**Figure 2.** Summary of mean relative contributions for sets of predictor variables. Summary of the mean relative contributions (%) ± standard deviation of different sets of predictor variables for boosted regression tree models for HPAI H5N1 (in blue) and H5Nx clade 2.3.4.4 (in red). The relative contribution is a measure of the relative importance of each predictor variable included in a BRT model to compute the model prediction. Set 1 predictor variables are represented on top, and Set 4 predictor variables are represented below.

Partial dependence plots of the BRT models allow the contribution of a particular variable to be depicted on the fitted response after taking into account the effect of all the other predictors in the model (*Figure 3*, and *Figure 3—figure supplement 1*). The main difference between the partial dependence plots of the different variables was for the density of extensively raised chickens, which showed a positive association with H5N1 HPAI presence contrasting with an absence of association with the H5Nx HPAI clade 2.3.4.4 presence (*Figure 3*). Other profiles were somewhat comparable for the two groups of viruses and showed a positive association between virus presence and duck density, intensively raised chicken density, human population density, a negative association with the IsChina variable (*Figure 3*) and an optimum for percentage of cropland and temperature (*Figure 3—figure supplement 1*). It should be kept in mind that their relative contributions, i.e. their weight in the final prediction strongly differed between the two groups of viruses. It is noteworthy that the models outlined above were built using optimal number of trees estimated through spatial CV instead of standard CV, and this resulted in much lower optimal number of trees compared to standard CV models (*Figure 3—figure supplement 2*), suggesting that standard CV may be over fitting local clusters of presence points rather than making reliable large-distance predictions. The suitability maps of the models are presented in *Figure 4*. To interpret the extrapolation capacity of these suitability maps, multivariate environmental similarity surfaces (MESS) (*Elith et al., 2010*) were computed (*Figure 4—figure supplement 3*) giving information on where the models extrapolate within the range of predictor variables in the occurrence points. As observed, both models extrapolate prediction in areas with similar environmental conditions, as depicted by positive MESS values.

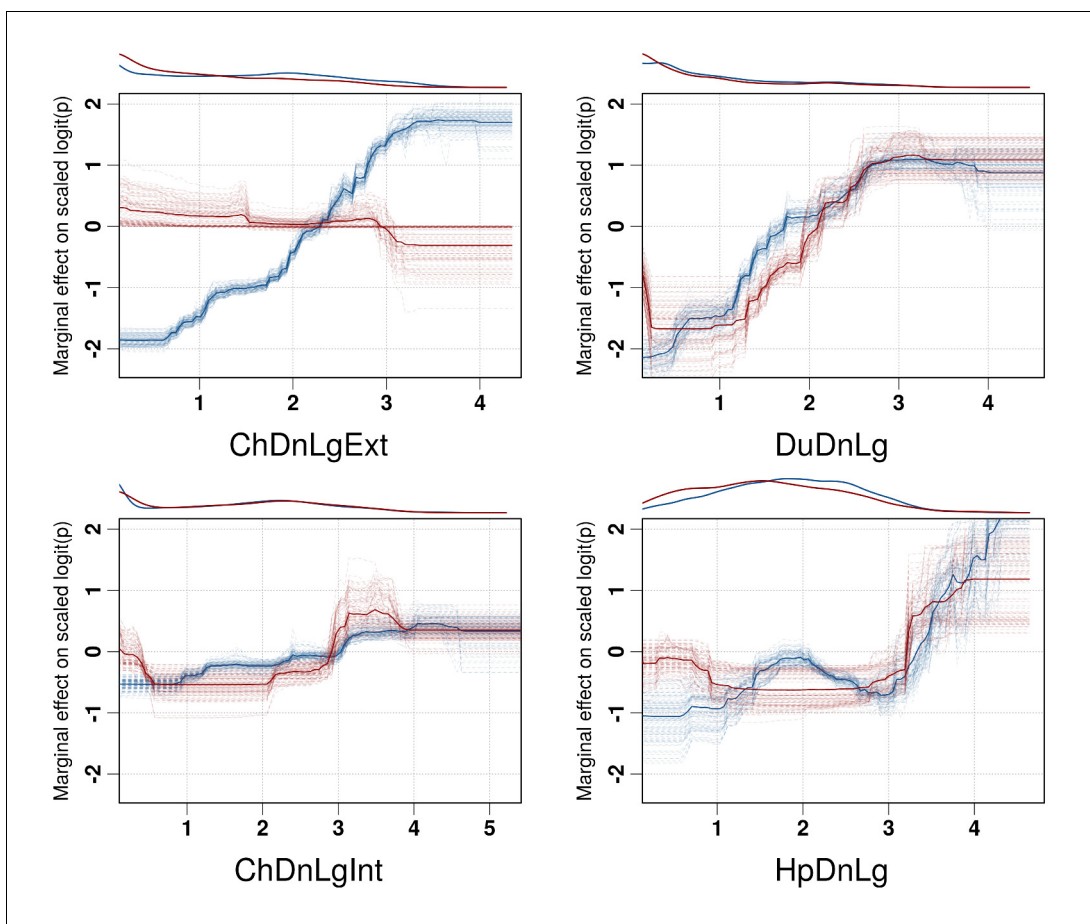

**Figure 3.** Boosted Regression Tree (BRT) profiles of selected predictor variables. BRT profiles or partial dependence plots of selected predictor variables for the global HPAI H5N1 (in blue) and H5Nx clade 2.3.4.4 model (in red). The BRT profiles provide a graphical description of the marginal effect of a predictor variable on the response (the probability of virus presence). The solid line represents the mean profile, whilst transparent lines represent each bootstrap. On the top of each plot, the density function of the observed distribution of predictors is displayed for one bootstrap and for the two datasets (HPAI H5N1- in blue and H5Nx clade 2.3.4.4- in red). Four predictor variables were selected for this figure: human population density (HpDnLg), extensive chicken density (ChDnLgExt), intensive chicken density (ChDnLgInt) and duck density (DuDnLg). The BRT profiles of Set 2, Set 3 and Set 4 predictor variables are represented in *Figure 3—figure supplement 1*. The optimal number of trees at which holdout deviance is minimised in the BRT models for all sets of predictor variables is represented in *Figure 3—figure supplement 2*.

The following figure supplements are available for figure 3:

**Figure supplement 1.** BRT profiles of Set 2, Set 3 and Set 4 predictor variables.

**Figure supplement 2.** Optimal number of trees at which holdout deviance is minimised in BRT models.

However, the geographical space with high similarity to the occurrence point is comparatively wider for the HPAI H5N1 model, than for the H5Nx clade 2.3.4.4 models.

As expected, high suitability values for the H5N1 HPAI model (*Figure 4*) are found in several parts of Asia, including China (when the effect of the IsChina variable is removed). Other areas where H5N1 HPAI has spread extensively are highlighted, such as eastern Indo-Gangetic plain, Thailand central plain, south Myanmar and the Red river and Mekong deltas of Vietnam, the island of Java in Indonesia and the Nile Delta in Egypt. The model also highlights areas where H5N1 HPAI was introduced but did not persist over long periods of time, such as in South Korea, Japan, Ukraine and

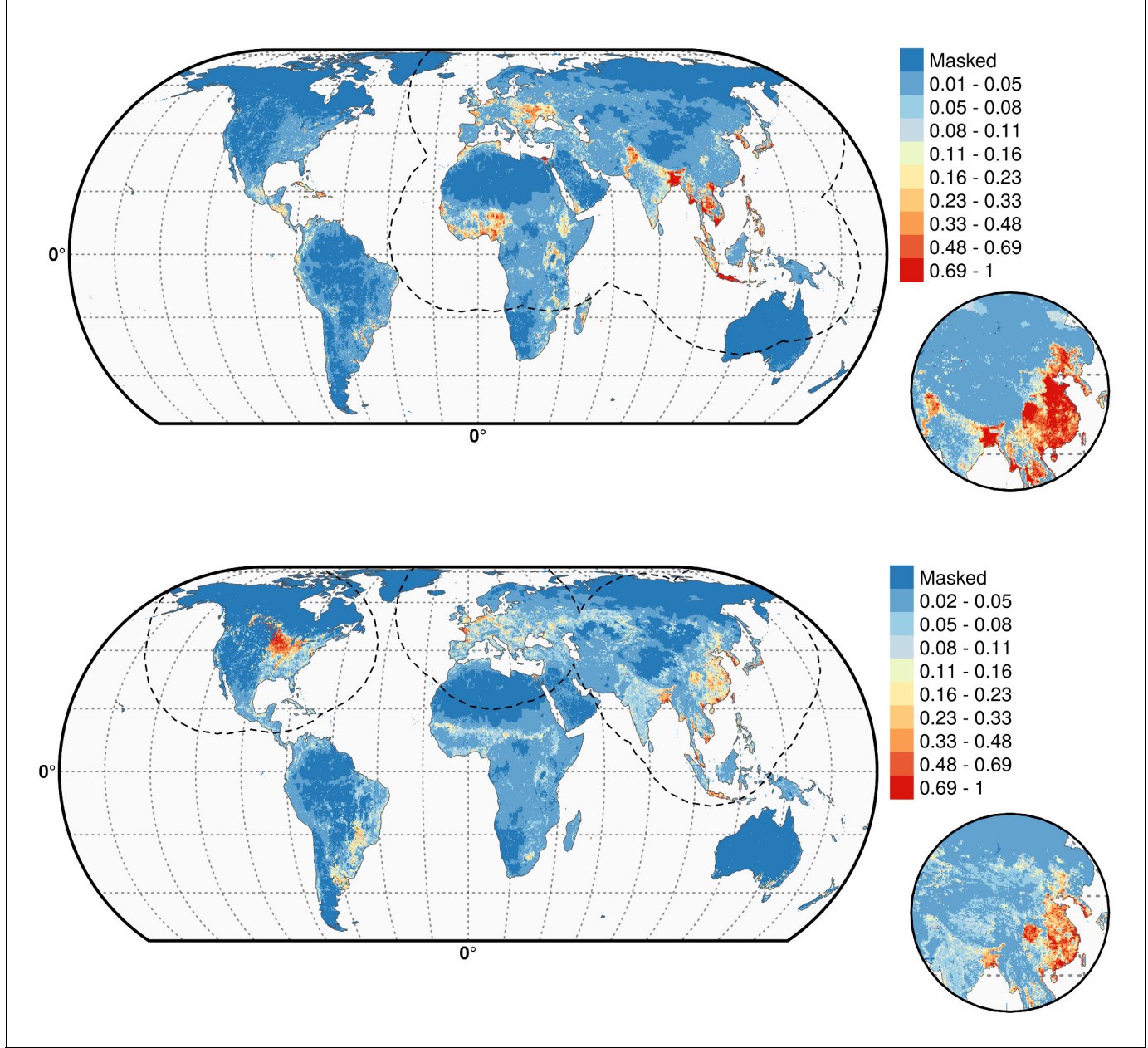

**Figure 4.** Predicted probability of occurrence of HPAI H5N1 and H5Nx clade 2.3.4.4. Predicted probability of occurrence of HPAI H5N1 for the Set 1 (top) and of H5Nx clade 2.3.4.4 for the Set 4 (bottom) (*Figure 4—source data 1* and *2* respectively). The dashed black line represents a buffer around the occurrence data for the HPAI H5N1 and H5Nx clade 2.3.4.4 predictions, corresponding to an area from which pseudo-absences were selected. The circle inset shows the prediction obtained when the effect of the variable IsChina was removed. The suitability maps HPAI H5N1 and H5Nx clade 2.3.4.4 for Set 2 and Set 3 variables are presented in *Figure 4—figure supplement 1* and *Figure 4—figure supplement 2* , respectively. The shapefile data used to produce these maps were all from public sources (http://www.naturalearthdata.com/). The graticule is composed of a 20-degree increments and the coordinate system is Eckert IV (EPSG: 54012). This figure was built with the R-3.2.4 software (https://cran.r-project.org/). Additionally, *Figure 4—figure supplement 3* depicts the Multivariate environmental similarity surfaces (MESS) maps for HPAI H5N1 and H5Nx clade 2.3.4.4 for the four sets of predictor variables.

The following source data and figure supplements are available for figure 4:

**Source data 1.** Suitability predictions for the HPAI H5N1 best model (GeoTiff format).

**Source data 2.** Suitability predictions for the H5Nx clade 2.3.4.4 best model (GeoTiff format).

*Figure 4 continued on next page*

*Figure 4 continued*

**Figure supplement 1.** Predicted probability of occurrence of HPAI H5N1 (top) and of H5Nx clade 2.3.4.4 (bottom) for the Set 2 variables.
**Figure supplement 2.** Predicted probability of occurrence of HPAI H5N1 (top) and of H5Nx clade 2.3.4.4 (bottom) for the Set 3 variables.
**Figure supplement 3.** Multivariate environmental similarity surfaces (MESS) maps for HPAI H5N1 and H5Nx clade 2.3.4.4.

Romania. Areas of western Africa, such as Nigeria, where the H5N1 HPAI outbreaks have been unfolding since late 2014 have been predicted as suitable by the model. Isolated parts in Eastern Europe, North America, Mexico, Dominican Republic and South America, are also deemed suitable for H5N1 establishment.

The suitability map for the H5Nx HPAI clade 2.3.4.4 virus is somewhat different, highlighting more isolated areas (*Figure 4*). The spatial extrapolation capacity of this model was low, and predictions made at large distances from known points of presence should be interpreted with caution. As this clade is still spreading, there may still be large areas of the landscape where it could potentially become established and where the model predictions may be inaccurate. In areas close to presence points, the predictions are believed to be robust, with several areas within Asia, such as China, South Korea, Japan and Taiwan depicted as suitable. The well-known virus 'reassortment-sink' areas of the Indo-Gangetic plains, the river deltas of Vietnam, southern Myanmar and Java, Indonesia, are also highlighted as areas of suitability. In Africa, the Nile delta is depicted as suitable for establishment. In North America, the high suitability areas match the intensive poultry areas of the Midwestern and southern states of USA. The Netherlands, Belgium and northwest France are highlighted with high suitability in Europe. In Australia, the commercial poultry rearing areas of Victoria and New South Wales are predicted as suitable; even though HPAI subtype H5 has never been reported in Australia.

The suitability predictions for the HPAI H5N1 and H5Nx clade 2.3.4.4 best models using Set 1 and Set 4 predictor variables are in *Figure 4—source data 1* and *2*, respectively. The suitability maps for H5Nx HPAI and H5 clade 2.3.4.4 for Sets 2 and 3 are presented in *Figure 4—figure supplement 1* and *Figure 4—figure supplement 2*, respectively.

## Discussion

A first important result of this study is that it was possible to build a global suitability model for HPAI H5N1 virus with a high extrapolation capacity robustly established through spatial cross-validation. Interestingly, H5N1 HPAI outbreaks appeared to be best modelled by predictor variables relating to host distribution. Alternative models based on land use or eco-climatic variables showed marginally better accuracy metrics when evaluated with standard CV, but significantly lower extrapolation capacity than the host-only variable. Even the models combining host variables with other environmental predictors did not produce significantly better results when evaluated through spatial CV. This observation matches earlier observations that association with eco-climatic variables were not consistently reproducible across countries and studies (*Gilbert and Pfeiffer, 2012*) and that H5N1 HPAI is probably not as strongly environmentally constrained as other authors have suggested (*Williams and Peterson, 2009*; *Zhang et al., 2014*). This strongly contrasts with vector-borne diseases, where clear eco-climatic boundaries of vectors can be mapped, and where climate has a strong influence on vector seasonality and population dynamics (*McMichael and Lindgren, 2011*; *Morin and Comrie, 2013*). In the case of a directly contagious disease such as avian influenza, successful transmission and clinical outbreaks have been observed over a wide range of temperature and humidity conditions (e.g. Russia, Nigeria, Egypt, Northern China, Indonesia). Our results suggest that the main large-scale constrains to suitability for H5N1 HPAI occurrence are related to the distribution of hosts; densities of chickens and ducks raised in different systems, and to the density of the human population, probably as a surrogate measure for various anthropogenic transmission mechanisms. For the new H5Nx clade 2.3.4.4 viruses, we found a somewhat different result, with a clear improvement of the extrapolation capacity of models using a set of variables combining host distribution and environmental variables. However, these models were of relatively low overall predictive power, most likely because the virus has not yet had a chance to extend fully to its potential range

of occurrence as compared to H5N1 HPAI, and false pseudo-absences may have had a strong impact on the construction of models and, therefore, on the accuracy of predictions. For this model, and given its low extrapolation capacity, we emphasise that predictions made at long distances from points of presence should be interpreted with caution, as there may still be large areas where it could potentially become established and where the our model predictions may be inaccurate.

A second important result of this study is to demonstrate the importance of spatial CV in building and validating avian influenza suitability models over large geographical extents. The difference between standard and spatial CV evaluation of GOF was already quite significant for the HPAI H5N1 models. The difference was even more striking for the H5Nx clade 2.3.4.4 models, which all appeared very good when evaluated through standard CV. However, they had poor extrapolation accuracy, sometimes even not much better than a null model when evaluated through spatial CV. Machine-learning techniques used in species distribution modelling have become incredibly powerful at reproducing a pattern from a given set of occurrence data and this is essentially what standard CV measures. However, as our results demonstrate, default cross-validation technique is a very misleading measure of their geographical extrapolation capacity. Spatial CV was found not only to be important for evaluating the extrapolation capacity of a given model, but also to be the only way to truly discriminate our model outputs based on different sets of predictor variables. The focus on extrapolation capacity for selecting predictor variables is driven by the assumption that a model that includes statistical relationships linked to causal mechanisms should spatially extrapolate well, as the cause-consequence statistical associations have a greater chance to apply well in different places than those that are coincidental. Of course, cause-consequence relationships may vary in space too, but the underlying assumption of suitability modelling extrapolation is that these remain constant over the spatial domain within which the model is applied. So, models based on coincidental statistical associations are expected to extrapolate poorly in the geographical domain, and these losses of predictability can hardly be quantified through standard CV because of spatial correlations between training and validation sets.

A third set of important results consisted in the comparison of the H5N1 HPAI and H5Nx clade 2.3.4.4 models, which showed areas of convergences and differences in the geographic and predictor variables spaces. Domestic duck density was the most important variable for both models, though with a lower RC for the H5Nx clade 2.3.4.4 model in Set 4. Ducks have always been strongly associated with areas of persistence and evolution of H5N1 HPAI (*Gilbert and Pfeiffer, 2012*), which relates to their capacity to act as an intermediate, domestic reservoir between wild *Anatidae*, the main wild reservoir of avian influenza viruses, and domesticated poultry. Ducks have been referred to as the 'Trojan horses' for H5N1 HPAI H5N1 presence (*Kim et al., 2014*) on account of their role in virus introduction, evolution, transmission and persistence (*Hulse-Post et al., 2005*), which has been demonstrated in both host pathogenicity (*Cornelissen et al., 2013*; *Smith and Donis, 2015*) and geospatial studies (*Gilbert and Pfeiffer, 2012*). The absence of duck density may in fact explain a lot of the difference in extrapolating capacity found between the host-model (Set 1) and the land-use and eco-climatic models (Set 2 and 3) that cannot discriminate areas with similar land-use and eco-climatic conditions, but that have very different duck densities. For example, India is predicted at relatively high suitability by the land-use model (Set 2, *Figure 4*) at a very low suitability by the host-based (except around Bangladesh) reflecting the near-absence of significant domestic duck densities in much of the country, in accordance with previous results (*Gilbert et al., 2010*).

The finding of a strong association between H5Nx clade 2.3.4.4 and ducks was somewhat less expected as the disease was found mostly in chicken farms in more intensive poultry production areas, but results are, however, in line with those of *Hill et al. (2015)* who found through phylogeographic analysis that the introduction of H5Nx clade 2.3.4.4 to South Korea was associated with areas where domestic ducks and wild waterfowl intermingled. Complex reassortment of multiple subtypes may also occur in areas where domestic ducks and migratory birds have an opportunity to share food, water and habitat, creating opportunities for virus transmission between different species, co-infection of individual animals with different influenza viruses and subsequent gene reassortment (*Deng et al., 2013*). It would be prudent for countries to put such areas under active surveillance for early detection of HPAI introductions and for monitoring of virus evolution. This would include the countries of the Americas and African continent where duck rearing is not as common as in South East Asia. It is noteworthy that one of the most severe recent H5 HPAI epidemics that started in 2015 in Dordogne region of France, a traditionally important duck rearing area with

some of the highest duck densities in the country, even if the outbreaks were apparently caused by distinct H5 viruses from those circulating in Asia. So, the association found with domestic duck densities fits with existing knowledge of H5N1 spatial epidemiology and was a major predictor in both the H5N1 HPAI and H5Nx clade 2.3.4.4 models.

In contrast, the association with extensively and intensively raised chickens provided different results for the H5N1 HPAI and H5Nx clade 2.3.4.4 models, with the latter being more strongly associated with intensified chicken production systems, found in intensive crop production areas with high human population densities. An interesting hypothesis to explain this pattern would be a greater fitness of H5Nx clade 2.3.4.4 viruses to spread through intensive chicken production and poultry trade systems (*Claes et al., 2016*). We still lack extensive published experimental infection results of the new clade in poultry, but preliminary results are indicative of a lower pathogenicity of the H5Nx clade 2.3.4.4 virus in chickens compared to H5N1 HPAI, with longer survival and shedding period (*Kim et al., 2014*; *Swayne et al., 2015*). A lower virulence in chicken was also found for the reassortant H5N2, H5N6 and H5N8 clade 2.3.4.4 viruses compared to previous 2.3.4 HPAI H5N1 viruses (*Sun et al., 2016*), although they remained highly pathogenic. A lower mortality and longer period of infectivity may assist the virus in circulating longer and within intensified poultry production and trading systems, leading to increased opportunities for onward transmission. Evolution towards reduced pathogenicity would appear an asset in improving farm-to-farm transmission and long-term persistence even in the absence of domestic ducks. This could partly explain the stronger association of H5Nx clade 2.3.4.4 viruses with intensive chicken production areas in eastern Asia and in the US.

Our analyses have focussed on poultry outbreak locations and are therefore of more limited use in identifying the locations of initial introduction of avian influenza viruses, or places where viruses may undergo more frequent reassortment events leading to the local emergence of new viruses. Future work may look more explicitly into those aspects and could lead to better prevention at the sources of virus introduction and emergence.

## Material and methods

### H5 location data

Two data sets corresponding to the two groups of viruses were compiled, respectively termed H5N1 HPAI, and H5 HPAI clade 2.3.4.4. The H5N1 HPAI data set was built from the database of the Global Animal Health Information System EMPRES-i of the FAO (*Food and Agriculture Organization of the United Nations, 2016*) (http://empres-i.fao.org/). A total of 17,068 confirmed outbreaks from January 2004 to March 2015 in poultry were used for this analysis, with the majority of outbreaks located in Asia, and no reports of H5N1 HPAI (not being of clade 2.3.4.4) in the Americas (*Figure 5*). In the absence of specific clade information on any given H5N1 HPAI outbreaks from 2013 onward, it was assumed to belong to the H5N1 HPAI data set (i.e. not being from clade 2.3.4.4). This may have resulted in some misclassification of some outbreaks in Eurasia, but their number relative to the total number of H5N1 HPAI outbreaks would be very low (<50) given the limited time period.

The H5 HPAI clade 2.3.4.4 data set was built by combining EMPRES-i outbreak location data with clade information from the Swiss Institute of Bioinformatics OpenFlu database (http://openflu.vital-it.ch/) using the procedure detailed in *Claes et al. (2014)*. In addition, searches on ProMed (http://www.promedmail.org/), the United States Department of Agriculture reports (http://www.usda.gov/avian_influenza.html), and other online literature were used for assignation of clade to H5 outbreaks. These included the H5N8, H5N2, H5N6, H5N3 and the recent H5N1 sequences from November 2013 to 15 June 2015. While this procedure was fairly straightforward for the newly emerged H5N8, H5N2, H5N6, H5N3 viruses, it was more challenging to assign a clade to the most recent H5N1 outbreaks. Hence, this H5 HPAI clade 2.3.4.4 data set only included those H5N1 outbreak records occurring after November 2013 that could be classified as clade 2.3.4.4, based upon documented evidence and confirmation from the above sources. This resulted in a dataset with 1309 outbreaks in poultry recorded as belonging to clade 2.3.4.4 from November 2013 to 15th June 2015 (*Figure 5*), involving 17 affected countries.

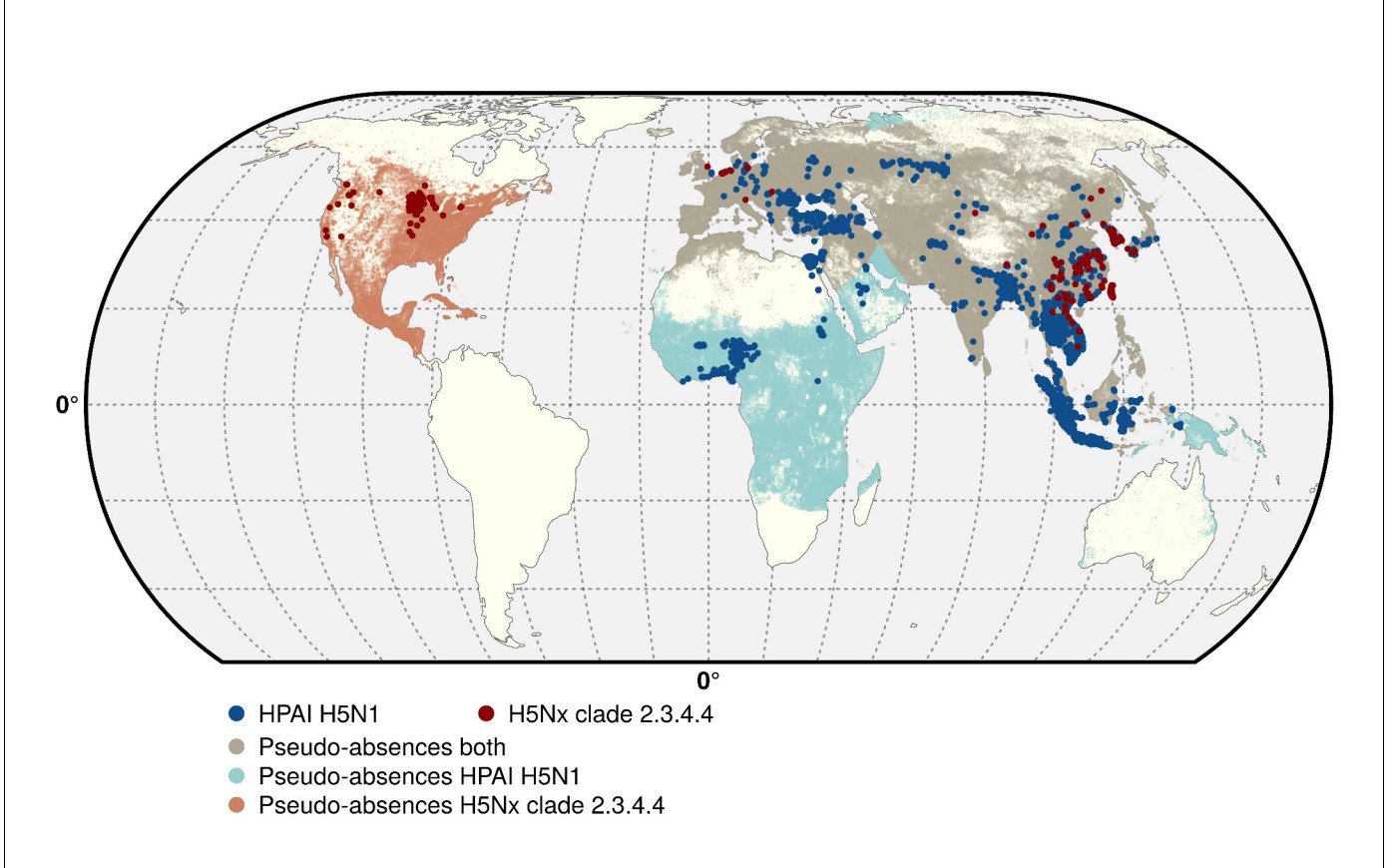

**Figure 5.** Geographic distribution of presence and pseudo-absences of HPAI H5N1 and HPAI H5Nx clade 2.3.4.4. Geographic distribution of presence points of HPAI H5N1 (blue) and HPAI H5Nx clade 2.3.4.4 (red). The pseudo-absences are represented in light blue, light red and light brown. This figure was built with the R-3.2.4 software (https://cran.r-project.org/). The shapefile data used to produce these maps were all from public sources (http://www.naturalearthdata.com/). The graticule is composed of a 20-degree increments and the coordinate system is 'EPSG: 54012'.

## Spatial predictor variables

Predictor variables traditionally associated with HPAI occurrence summarised in a recent literature reviews (*Gilbert and Pfeiffer, 2012*) were selected in addition to a few others. Three categories of variables were included: hosts, land use/land cover and eco-climatic variables. Host variables included log10-transformed extensive and intensive chicken density (*Gilbert et al., 2015*), duck density (*Robinson et al., 2014*) and human population density. Whilst the poultry variables were available as global databases (with the exception of ducks, which were computed as detailed below), the human population density layer was built from two different data sources; the Worldpop database (http://www.worldpop.org) in all countries where it was available across Africa (*Linard et al., 2012*), Asia (*Gaughan et al., 2013*) and South America (*Sorichetta et al., 2015*) and the Center for International Earth Science Information Network's Gridded Population of the World (GPW) database elsewhere (*Socioeconomic Data and Applications Center, 2016*) (http://sedac.ciesin.columbia.edu/entri). Since both data sets are standardised to match UN national totals, these two databases should be consistent against each other.

The Global Duck Distribution Data were computed using The Gridded Livestock of the World (GLW) version 2 (http://livestock.geo-wiki.org), which only included duck data on Asia, Europe and North America. Using the GLW downscaling method and spatial predictors presented in *Robinson et al. (2014)*, we developed a global-scale model of duck distribution at a spatial resolution of 0.083333 decimal degrees resolution, using all global data available to date on duck distribution in the FAO Global Livestock Information System (GLIS). These new modelled values were used

to avail predicted values in Africa and South America, whilst the original GLW version two predictions were maintained for the continents where they were available.

For the land cover data, we used the Global 1 km Consensus Land Cover database (*Tuanmu and Jetz, 2014*) that distinguishes land use and land cover classes (*Table 1*) with an index of the prevalence of each class in percentage for a ~1 km pixel (http://www.earthenv.org/landcover.html). These data layers were supplemented by a layer about the distance of each spatial point to the open water. Finally, a third set of spatial predictors (*Table 1*) describing the seasonality and large-scale pattern of eco-climatic indices such as day-time land surface temperature (day LST) and Normalised Difference Vegetation Index (NDVI) was also used (*Scharlemann et al., 2008*).

Finally, an additional covariate to account for mass vaccination of poultry against H5N1 in China was also included (IsChina), which could not be captured by any other predictor variable. This term was added only for China because the role played by mass vaccination is believed to be much higher than in any other countries. Post vaccination seropositivity in China ranges between 80% and 95% in China (*Martin et al., 2011*), whereas papers having looked at post-vaccination seropositivity in Indonesia (*Sawitri et al., 2007*), Vietnam (*Domenech et al., 2009*) and Egypt (*Rijks and ElMasry, 2009*) found it to be insufficient to successfully prevent transmission, often close to 30%. China is also by far the biggest user of vaccines: '125 billion doses of H5N1 vaccine were produced and deployed in total; in China (120 billion), Indonesia (3 billion) and Viet Nam (2 billion) between 2004 and 2012'(*Castellan et al., 2014*). All the risk factor variables were at a spatial resolution of 0.083333 decimal degrees per pixel, which equals an approximate resolution of 10 by 10 km at the equator.

The predictor variables were categorised into four sets to predict the probability of virus presence (*Table 1*). Set 1 included the host variables of extensively (ChDnLgExt) and intensively raised chicken density (ChDnLgInt), duck density (DuDnLg), human population density (HpDnLg) and the effect of mass vaccination in China (IsChina). Set 2 included the land use and land cover variables and IsChina, whereas Set 3 included all eco-climatic variables and IsChina. Finally, Set 4 included a selection of variables from the earlier sets that were selected on the basis of prior epidemiological knowledge (*Gilbert and Pfeiffer, 2012*). These included all variables from Set 1 in addition to (i) the land cover 'Cultivated and Managed Vegetation' class accounting for the association between poultry and cropping patterns, (ii) the land cover 'Open Water' and 'Distance to water' class accounting for the persistence of the virus in landscapes rich in water environment, variables previously found associated with H5N1 HPAI presence in China (*Shaman and Kohn, 2009*), (iii) the day LST annual mean to account for the persistence of virus in the environment which has been shown to vary with temperature (*Liu et al., 2007*; *Zhang et al., 2014*). The combination of variables from Set 1 and Set 2 on one hand (Set 2.1), and of Set 1 and Set 3 (Set 3.1) on the other hand were also investigated.

## Modelling procedure

Boosted Regression Tree (BRT) models (*Elith et al., 2006*) were employed to predict the probability of occurrence of H5N1 HPAI viruses and H5Nx HPAI clade 2.3.4.4, as a function of the sampled predictor variables. We used BRT as it allows for modelling of complex non-linear relationships to be modelled using various types of predictor data and takes into account the interactions between predictor variables (*Elith et al., 2008*). BRT models generate a large number of regression trees, fitted in a stepwise manner, for optimising the predictive probability of occurrence based on predictor variable values, as compared to several other modelling methods (*Elith et al., 2006*) and has been shown to produce accurate predictions of H5N1 (*Martin et al., 2011*) and H7N9 subtypes (*Gilbert et al., 2014*).

BRT models require data on both presence (provided by two H5 data sets) and absence, and we modelled two separate outcomes using the parameters described further in the section; the presence/absence of H5N1 HPAI and H5Nx HPAI clade 2.3.4.4 viruses. Whilst presence is derived from the two respective H5 data sets, absence data are rarely measured through active surveillance, so need to be approximated by generating pseudo-absences points. The literature yields no consensus on the correct approach to generate pseudo-absence data, so we used an evidence-based probabilistic framework for generating pseudo-absence data points incorporating the main biasing that may have affected the distribution of the presence points (*Phillips et al., 2009*). We used the bgSample function from the 'seegSDM' package (https://github.com/SEEG-Oxford/seegSDM) (*Phillips et al., 2009*; *Pigott et al., 2014*) to generate a pixel level spatial distribution of pseudo-absence based on the human population distribution (*Figure 5*) to account for differences in surveillance and reporting

intensity. This was based on the assumption that under-reporting would be more likely in remote areas with low population density than in highly populated, where the disposal or dead birds and carcasses would more hardly go unnoticed. In addition, the Empres-i database compiles outbreak locations data from very heterogeneous sources and in the absence of explicit GPS location data, the geo-referencing of individual cases is often through the use of place name gazetteers that will tend to force the outbreak location populated place, rather in the exact location of the farm where the disease was found, which would introduce a bias correlated with human population density. Finally, this also allowed to prevent any pseudo-absences in unpopulated regions.

With dynamically spreading pathogens, 'absences' may result from a genuine unsuitability for infection, a lack of surveillance or reporting while the pathogen is present, or simply the fact that the pathogen has not been introduced to a region. Minimum and maximum distances to the nearest presence observations were therefore introduced in the selection of pseudo-absence points to limit that effect (*Phillips et al., 2009*). The minimum distance was set to 10 km in both models, in relation to outbreak surveillance zones for HPAI in most countries. The maximum distance to the nearest positive observation could not be informed by surveillance strategies and was randomly set between 1000 and 3000 km across different bootstrapped runs of the model in order to ensure that the results were not too sensitive to a specific maximum distance. Prior to running the model, the duplicate points falling in the same pixel were summarised, in order to label each pixel as 'presence' or 'pseudo-absence'. This procedure resulted in a reduced data set with 5038 and 403 presence points (pixels) for the H5N1 HPAI and H5Nx HPAI clade 2.3.4.4 models, respectively (*Figure 5*).

To select the optimal number of trees in the BRT models, the k-fold cross validation procedure described in *Elith et al. (2008)* was employed, using the R package dismo. Each model was run with four different sets of predictor variables to measure their respective predictive power. In addition, the weight of each predictor variable was also evaluated individually by their relative contribution, a metric was produced that described the proportion of times a particular variable was selected by the model for splitting a decision tree, and the overall improvement it brought to the model (*Friedman and Meulman, 2003*). In addition to the standard random cross-validation procedure of *Elith et al., (2008)*, a calibrated cross-validation was also computed to account for the SSB (*Hijmans, 2012*). Clustering of occurrences in species distribution models may lead to inflation of cross-validation metrics because the distance between training-presence and testing-presence sites will tend to be smaller than the distance between training-presence and testing-absence sites (referred to as SSB). To account for SSB, the testing data were sub-sampled using the distance to training data. The first step in this approach is to compute, for each testing-presence site, the distance to the nearest training-presence site. During the sub-sampling procedure, each testing-presence site is paired with the testing-absence site that has the most similar distance to its nearest training-presence site. If the difference between the two distances is more than a specified threshold (33%) the presence site is not used. This procedure ensures that clustering of presence data is accounted for and avoids the inflation of model evaluation metrics.

In addition, we implemented spatial CV, whereby training and testing sets are partitioned on a spatial basis, in order to quantify how model predictions could extrapolate geographically (*Gilbert et al., 2014*; *Randin et al., 2006*; *Wenger and Olden, 2012*). Disease outbreak data are typically clustered, or spatially autocorrelated, and this may bias standard cross validation (CV) procedures because the training and validation data sets are not independent from each other (*Randin et al., 2006*). A possible consequence is that the goodness of fit metrics provided by the standard CV procedure may overestimate the real capacity of the model to make reliable predictions in areas distant from the training set. The spatial CV procedure was performed by partitioning non randomly the study area into five spatial clusters (*Figure 6*) by first selecting five reference presence points. A minimum distance was specified between the selected points to obtain a balanced sample size between the clusters. These selected points represent the benchmarks to build the five-folds/clusters of the spatial CV models. Thereafter, the nearest benchmark presence to each observation is identified and labelled with this benchmark point. Finally, the five clusters containing presences and absences are delineated and are used as folds in the spatial cross validation procedure. In the procedure described by *Elith et al. (2008)*, an optimal number of trees for the BRT model is found by finding the minimum deviance to the evaluation set. By replacing the standard CV by the spatial CV, we also allow the optimal number of trees to correspond to the minimum deviance in a geographically distant evaluation set. Both the BRT models were run with the following parameters; a

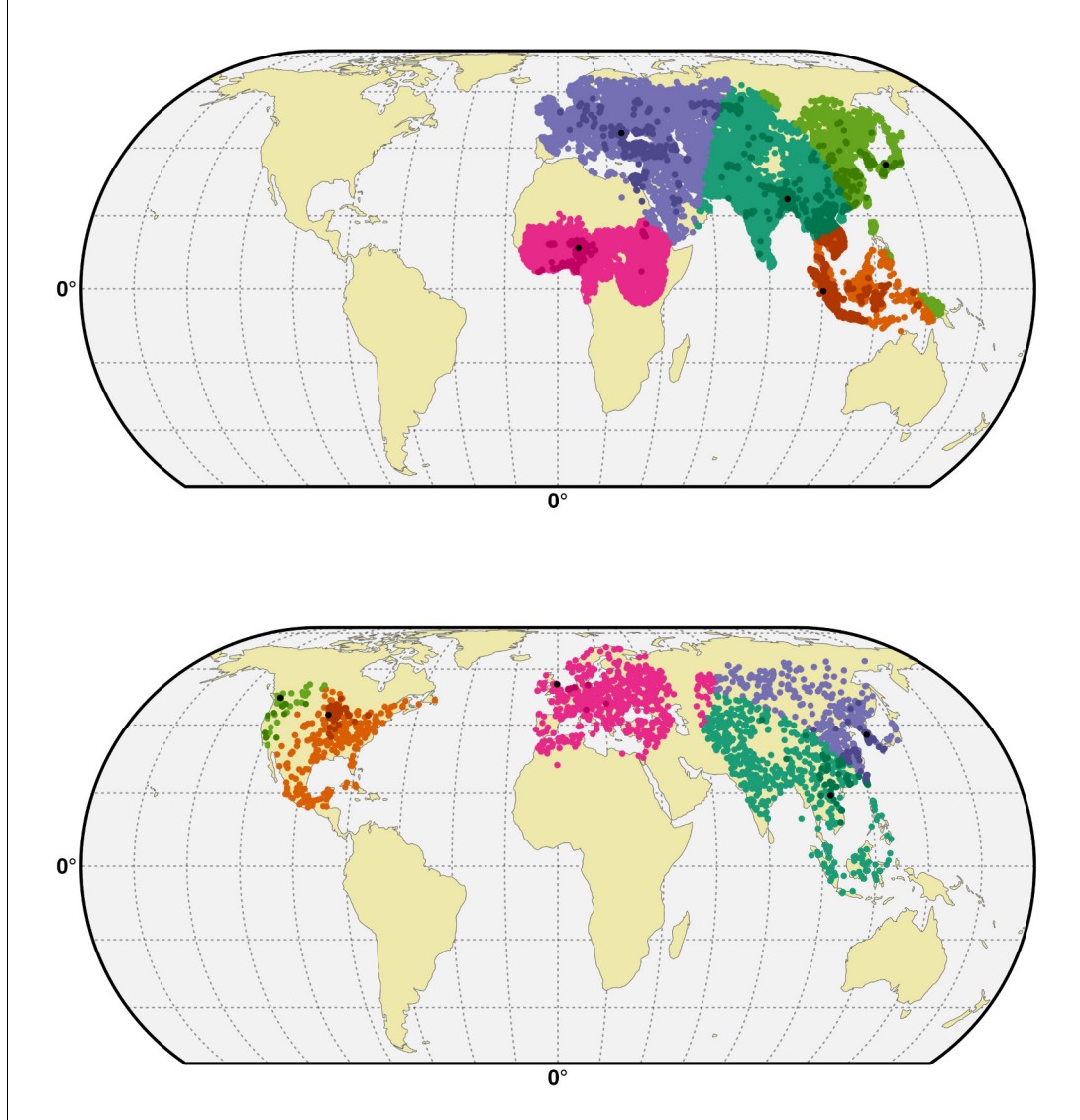

**Figure 6.** Spatial cross-validation partition for H5N1 HPAI and H5Nx clade 2.3.4.4. Visualisation of a typical partition used for the spatial cross-validation of the H5N1 HPAI (top) and H5Nx clade 2.3.4.4 (bottom). The presence and pseudo-absences are partitioned into *k* (five) clusters for training and testing set. One cluster is used for testing data and *k-1* clusters are used for sampling training data. The *k* (five) reference presence points (randomly sampled in each bootstrap) used to build each clusters are represented in black in the map. The code used for implementing the spatial cross validation is detailed in *Source code 1*. This figure was built with the R-3.2.4 software (https://cran.r-project.org/). The shapefile data used to produce these maps were all from public sources (http://www.naturalearthdata.com/). The graticule is composed of a 20-degree increments and the coordinate system is 'EPSG:54012.

tree complexity of 4, and the initial number of trees set at 100. For the HPAI H5N1 and clade H5Nx 2.3.4.4 model, a learning rate of 0.01 and 0.005, respectively, was used. A step size of 200 and 50 trees was used for the HPAI H5N1 and clade H5Nx 2.3.4.4 models, respectively.

All BRT models were bootstrapped across 20 values of maximum distance to the presence point. For each set of parameters, five splits of training and testing dataset were randomly sampled to compute the CV metrics. All in all, 100 bootstraps were used per group of viruses and per set of predictor variables. The GOF of the models was calculated using Area Under the Receiver Operating Curve (AUC) metrics, and the mean predictions from the bootstrapped models were generated on a continuous scale of 0 to 1 for each pixel, to be mapped over the study area.

We produced the predictions on a global scale to predict the global suitability of H5N1 HPAI and H5Nx clade 2.3.4.4 presence. There are certain uncertainties associated with extrapolating over large geographical domains, and hence, in order to delimit the environments outside of the range of model calibration locations, the multivariate environmental similarity surfaces (MESS) (*Elith et al., 2010*) was computed on each set of predictors and set of occurrence points.

## Acknowledgements

TPR is funded by the ESEI UrbanZoo (G1100783/1), BBSRC-ZELS ZooLinK (BB/L019019/1) programs, and is further supported by the Consultative Group on International Agricultural Research (CGIAR) Research Programs on Agriculture for Nutrition and Health (A4NH) and Livestock. Part of this work was funded by the US National Institutes of Health (1R01AI101028-02A1), United States Agency for International Development (USAID) Emerging Pandemic Threats program, and by the FNRS project 'Mapping people and livestock' (PDR T.0073.13)

## Additional information

### Funding

| Funder | Grant reference number | Author |
|---|---|---|
| National Institutes of Health | 1R01AI101028-02A1 | Madhur S Dhingra<br>Jean Artois<br>Xiangming Xiao<br>Marius Gilbert |
| Biotechnology and Biological Sciences Research Council | BB/L019019/1 | Timothy P Robinson |
| Medical Research Council | ESEI UrbanZoo (G1100783/1) | Timothy P Robinson |
| CGIAR | Research Programs on Agriculture for Nutrition and Health (A4NH) and Livestock | Timothy P Robinson |
| Fonds De La Recherche Scientifique - FNRS | PDR T.0073.13 | Catherine Linard<br>Marius Gilbert |
| United States Agency for International Development | Emerging Pandemic Threats program | Scott H Newman |

The funders had no role in study design, data collection and interpretation, or the decision to submit the work for publication.

### Author contributions

MSD, JA, Acquisition of data, Analysis and interpretation of data, Drafting or revising the article; TPR, FC, SHN, Conception and design, Drafting or revising the article; CL, XX, Acquisition of data, Drafting or revising the article; CC, RE, Analysis and interpretation of data, Drafting or revising the article; IX, Conception and design, Drafting or revising the article, Contributed unpublished essential data or reagents; RL, DK, Drafting or revising the article, Contributed unpublished essential data or reagents; SVD, GD, Conception and design, Acquisition of data, Drafting or revising the article; MG, Conception and design, Acquisition of data, Analysis and interpretation of data, Drafting or revising the article

### Author ORCIDs

Timothy P Robinson, http://orcid.org/0000-0002-4266-963X
Marius Gilbert, http://orcid.org/0000-0003-3708-3359

## Additional files

**Supplementary files**
• Source code 1. R script implementing the cross validations (CV); namely, partition into geographic folds, running the BRT models with standard CV, standard CV accounting for the spatial sorting bias (SSB), and the spatial CV.

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
