## [Decision Letter]

Thank you for submitting your article "Global mapping of highly pathogenic avian influenza H5N1 and H5Nx clade 2.3.4.4 viruses with spatial cross-validation" for consideration by *eLife*. Your article has been favorably evaluated by Wendy Garrett (Senior Editor) and three reviewers, one of whom served as a guest Reviewing Editor. One of the three reviewers, Richard J Webby, has agreed to share his name.

The reviewers have discussed the reviews with one another and the Reviewing Editor has drafted this decision to help you prepare a revised submission.

Summary:

The manuscript describes the development of global risk maps for H5N1 and H5NX viruses. Building on previous work mapping risk by country and by region, the authors sought to build parsimonious global suitability models based on host and environmental factors. In order to discriminate among models incorporating different variables the authors used boosted regression trees and identified human and animal population densities as key factors influencing risk (as supported by previous work). While the basic factors identified are perhaps not unexpected, refinement and improvement to mapping practices are needed to help support global preparedness and response.

All reviewers agree that the data shed light on an important and interesting problem. However, before we can recommend acceptance, a number of issues should be addressed.

Essential revisions:

1) It is not clear why the distinction was made between clade 2.3.4.4 viruses and all H5 viruses others. As pointed out, the 2.3.3.4 viruses did have different neuraminidases and did transmit to North America, but there is no convincing evidence that we are aware of that supports the authors assumption that "they may have different phenotypic characteristics leading to different transmission patterns". There are very likely different risk factors combined in the "other clade" category. For example, clade 2.2 and clade 2.3.2.1 viruses spread out of Asia much like 2.3.4.4 (with obvious exception of migration to Americas) and at least appeared more associated with wild bird spread. Were any other subclade analyses done? Is there evidence, or at the least practical reason, to support clumping all other clades together?

2) The use of human population distribution for pseudo-absence generation needs to be better justified. Given that the purpose of pseudo-absence generation is to reflect the bias in sampling or reporting effort present, it is not obvious that human population density best reflects this difference. We suggest the authors consider approaches that integrate other strains of avian influenza as direct absence locations. Given EMPRESi as a data source, this should at least be feasible at least for H7 viruses. Otherwise, the authors need to provide more context for why such testing would scale with human population; it could be argued that testing could scale with avian distributions better than it does humans (e.g. in the United States). There may also exist other, more appropriate, high resolution proxies for this reporting bias.

3) As pointed out in the Methods section, the inference of pseudo-absences can be biased by the fact that the pathogen has not been introduced into a region. This is undoubtedly true but the assertion that this bias is likely to be stronger for H5Nx than H5N1 viruses is peculiar. The suitability maps in Figure 4 differ substantially for North America, presumably because of the lack of H5N1 infections and thus inferred absence. However, as in comment 1 above, we are unaware of evidence to suggest that there are phenotypic differences between the H5N1 and H5Nx viruses that would make North America suitable for H5Nx viruses but unsuitable for H5N1 viruses. It could be argued that H5N1 has also not yet had the opportunity to explore its full potential range.

4) In the sixth paragraph of the Results the authors discuss the results of Chinas inclusion in the high suitability zone, but essentially only when the IsChina variable was removed. What exactly does this mean regarding the influence of this variable? It is obviously clear China is very much a high suitability zone for these viruses. Along similar lines, if IsChina is meant to account for mass poultry vaccination in China, why have similar variables been ignored for other countries that have or have had mass vaccination campaigns – particularly Indonesia, Egypt, and Vietnam?

5) The authors produce predictions that are many miles away from any previously reported occurrences – whilst the dotted lines on Figure 4 help the reader be aware of this, the authors should also consider a multivariate environmental similarity surface approach to gauge the extent to which these records are extrapolation, and where predicted values fall within environmental space already considered by the presence points already in place. There is an example of its usage in Elith et al. 2010 http://onlinelibrary.wiley.com/doi/10.1111/j.2041-210X.2010.00036.x/full

6) We are concerned about the independence of the occurrence points, particularly with respect to secondary spread rather than an inherent feature of the underlying environment in these locations. An example of where this is problematic would be charting the progress of Nipah outbreaks in Malaysia in 1998 – pigs and humans over a widespread area were infected yet in terms of environmental suitability, only one location, where the index case arose, is suitable for inclusion. All other cases are more appropriately modelled by understanding the nature of swine connectivity via livestock trade and human-animal interactions. To what extent are these influenza cases located, and to what extent are they environmentally driven?

7) This exposes a broader issue relating to the inclusion of the poultry layers in the model. Niche maps are simply reflections on the potential for presence of the pathogen – by using data from poultry and then including poultry measures as predictors it is therefore unsurprising that they have a high explanatory power within the model, particularly where there may be ineffective control for the biasing effects this may have. As a consequence, the maps may be a more accurate prediction of where outbreaks are more likely to occur (or be reported) rather than an accurate depiction of potential occurrence (i.e. anything from singular cases to very many). It was interesting therefore to see how the inclusion/exclusion of different covariates impacted this – even exploring what impact that has when converted into a map would be interesting to see, particularly an output based just upon Sets 2 and 3 alone.

[Editors' note: further revisions were requested prior to acceptance, as described below.]

Thank you for resubmitting your work entitled "Global mapping of highly pathogenic avian influenza H5N1 and H5Nx clade 2.3.4.4 viruses with spatial cross-validation" for further consideration at *eLife*. Your revised article has been favorably evaluated by Wendy Garrett (Senior editor), a Reviewing Editor, and one of the original reviewers.

The manuscript has been improved but there are some remaining issues that need to be addressed before acceptance, as outlined below:

*Reviewer #2:*

The authors response to reviewer comments have clarified some initial concerns as well as provided interesting additional analyses. However, I do have some still outstanding issues, as well as some newly raised concerns given the authors response.

It was interesting to note the use of gazetteers to approximate disease occurrence since this adds an additional concern as to how reliable this classification is, and if in doubt, what steps were taken to mitigate (e.g. turning specific lat longs into larger polygons capturing such uncertainty).

The authors state that different pseudo-absence models produced "similar results", but provide no means for reviewers to gauge that similarity.

The clarification the authors present that these outputs are depicting suitability for infection, including secondary spread, to my mind is problematic, since this requires mixing different types of data together within the same model. It is therefore unclear to me how the disparate effects of environment (likely influencing primary infection) and human/movement/transmission networks (likely influencing secondary infection) will be disaggregated. Whilst the cross-validation may mitigate the potential clustering that primary and secondary cases have, I'm still not 100% convinced that this differing process can be adequately captured simultaneously. To my mind a more powerful demonstration of difference (or indeed lack of) would be a comparison of primary and secondary data alone.

It was good to see the inclusion of Set 2 and Set 3 results – what was disappointing however was that there was no discussion of the differences this produces (sometimes considerable e.g. India for HPAI). How much of this variation is due to differences in ratios of primary v. secondary cases considered as occurrence records across the different geographies for instance? I found it very interesting to see such a different picture result when the human/host variables were dropped.

---

## [Author Response]

*1) It is not clear why the distinction was made between clade 2.3.4.4 viruses and all H5 viruses others. As pointed out, the 2.3.3.4 viruses did have different neuraminidases and did transmit to North America, but there is no convincing evidence that we are aware of that supports the authors assumption that "they may have different phenotypic characteristics leading to different transmission patterns". There are very likely different risk factors combined in the "other clade" category. For example, clade 2.2 and clade 2.3.2.1 viruses spread out of Asia much like 2.3.4.4 (with obvious exception of migration to Americas) and at least appeared more associated with wild bird spread. Were any other subclade analyses done? Is there evidence, or at the least practical reason, to support clumping all other clades together?*

There were a few recent virology studies that suggest milder clinical disease in duck favoring long-distance transmission (e.g. DeJesus et al. 2016 doi:10.1016/j.virol.2016.08.036), but we agree that this statement of possible different phenotypic characteristics is a bit too speculative to justify the distinction. Making a full clade-level analysis would go beyond the scope of the present study, and would require in-depth data mining to be able to allocate clades to all known outbreaks. This is an approach we recently followed, over a much more limited geographical domain (Artois et al. 2016 doi 10.1038/srep30316).

However, the very fast spread of clade 2.3.4.4 H5Nx viruses and the fact that it involved multiple reassortments leading to many different viruses with different neuraminidase is quite unprecedented (de Vries et al. 2015 doi 10.3201/eid2105.141927), and warranted further examination of how it had spread in the geographical and environmental space, in comparison to previous known HPAI H5N1 viruses.

So, we removed the most speculative statements and tried to better justify this grouping based on the above argument. We also somewhat restructured the Introduction to make it clearer that the main focus of the paper is on the spatial epidemiology of HPAI H5N1 in general, and that the analysis of the 2.3.4.4 H5Nx viruses comes as a comparative addition.

*2) The use of human population distribution for pseudo-absence generation needs to be better justified. Given that the purpose of pseudo-absence generation is to reflect the bias in sampling or reporting effort present, it is not obvious that human population density best reflects this difference. We suggest the authors consider approaches that integrate other strains of avian influenza as direct absence locations. Given EMPRESi as a data source, this should at least be feasible at least for H7 viruses. Otherwise, the authors need to provide more context for why such testing would scale with human population; it could be argued that testing could scale with avian distributions better than it does humans (e.g. in the United States). There may also exist other, more appropriate, high resolution proxies for this reporting bias.*

This choice was made for several reasons. First, in a context of underreporting as happening in much of Asia, we assumed that under-reporting would be easier in remote areas with low population density than in regions with high population density, where the disposal or dead birds and carcasses would be less likely to go unnoticed. Second, the Empres-I database compiles outbreak locations data from very heterogeneous sources and the geo-referencing of individual cases, in the absence of explicit X-Y locations, is often made through place names gazetteers that will tend to place outbreak location in the closest village, or administrative unit main village rather in the exact location of the farm where the disease was found, which will introduce a bias correlated with human population density.

Third, in much of Asia, rural population density is somewhat correlated with poultry density. However, we agree that this needed to be established more carefully, so we re-ran our models using poultry density instead of human population density to generate pseudo-absences, and we obtained very similar results.

We looked into the possibility of using H7 viruses, but the large majority of H7 records in Empres-I are in fact H7N9 positive samples from markets in China (Gilbert et al. 2014, doi 10.1038/ncomms5116), which are concentrated within cities. So, we felt that this would have the risk of introducing an unwanted bias.

The choice of human population density for pseudo-absence generation is now better justified in the manuscript (added in subsection “Modelling procedure”.)

*3) As pointed out in the Methods section, the inference of pseudo-absences can be biased by the fact that the pathogen has not been introduced into a region. This is undoubtedly true but the assertion that this bias is likely to be stronger for H5Nx than H5N1 viruses is peculiar. The suitability maps in Figure 4 differ substantially for North America, presumably because of the lack of H5N1 infections and thus inferred absence. However, as in comment 1 above, we are unaware of evidence to suggest that there are phenotypic differences between the H5N1 and H5Nx viruses that would make North America suitable for H5Nx viruses but unsuitable for H5N1 viruses. It could be argued that H5N1 has also not yet had the opportunity to explore its full potential range.*

We did not infer absence of HPAI H5N1 in North America, as our pseudo- absences were filtered by geographic distance from positives (as indicated in the Materials and methods), which prevented any pseudo-absence to fall with the Americas. So, the absence of significant suitability for HPAI H5N1 is not related to inferred absences there. We agree, however, that we can hardly exclude that H5N1 could do well in North America, but not necessarily so. One should keep in mind that HPAI H5N1 was seeded throughout Western Europe in several countries and locations and never turned into a real epidemic. Domestic duck density has a relatively high relative contribution in our models, and was shown in many previous researches to be associated with HPAI H5N1 spread in Asia at multiple spatial scale. The absence of high domestic duck densities may then partly explain the predicted low suitability in the Americas.

We removed the statement suggesting that this effect may be stronger for the H5Nx in the Materials and methods section and now only comment the study results, i.e. that the extrapolation capacity quantified through spatial cross validation was found to be lower for the H5Nx viruses. In addition, we updated the figure with pseudo-absences to better visualize the respective sets of pseudo- absences used for each model (HPAI H5N1 and H5Nx clade 2.3.4.4).

*4) In the sixth paragraph of the Results the authors discuss the results of Chinas inclusion in the high suitability zone, but essentially only when the IsChina variable was removed. What exactly does this mean regarding the influence of this variable? It is obviously clear China is very much a high suitability zone for these viruses. Along similar lines, if IsChina is meant to account for mass poultry vaccination in China, why have similar variables been ignored for other countries that have or have had mass vaccination campaigns – particularly Indonesia, Egypt, and Vietnam?*

China is very much in the high suitability zone for these viruses. To clarify, even with the inclusion of the IsChina variable, large parts of China still remain suitable for HPAI H5N1. But we have strong reasons to believe that the effect of mass-vaccination is far more pronounced in China than in these other countries, which justifies using this term in our models. Post vaccination seropositivity in China ranges between 80 and 95% in China (see Figure 5 in Martin et al. 2011 doi: 10.1371/journal.ppat.1001308), whereas papers having looked at post- vaccination seropositivity in Indonesia (Sawitri et al. 2007), Vietnam (Domenech et al. 2009), and Egypt (Rijks 2009) found it to be insufficient, often close to 30%. China is by far also the biggest user of vaccines: “125 billion doses of H5N1 vaccine were produced and deployed in total; in China (120 billion), Indonesia (3 billion), and Viet Nam (2 billion) between 2004 and 2012” (Castellan et al., 2014). This justification has been added to the manuscript (subsection “Spatial predictor variables”, fourth paragraph).

Sawitri, E. S., Darminto, J. Weaver, and A. Bouma. The vaccination program in Indonesia. In: Vaccination: a tool for the control of avian influenza, Proc. of the Joint OIE/FAO/IZSVe Conference, Verona, Italy March 2007. Istituto Zooprofilattico Sperimentale delle Venezie, Dodet and the Scientific and Technical Department of the O.I.E., eds. 130:151– 158.Karger, Basel, Switzerland. 2007.

Domenech, J., G. Dauphin, J. Rushton, J. McGrane, J. Lubroth, A. Tripodi, J. Gilbert, and L. D. Sims. Experiences with vaccination in countries endemically infected with highly pathogenic avian influenza: the Food and Agriculture Organization perspective. O.I.E. Rev. Sci. Tech. 28:293– 305. 2009.

Rijks J, ElMasry I. Food and Agriculture Organisation report. 2009. Characteristics of poultry production in Egyptian villages and their effect on HPAI vaccination campaign results – Results of a participatory epidemiology study.

Castellan, D.M., Hinrichs, J., Fusheng, G., Sawitri, E., Dung, D.H., Martin, V., McGrane, J., Bandyopadhayay, S., Inui, K., Yamage, M., Ahmed, G.M., Macfarlane, L., Williams, T., Dissanayake, R., Akram, M., Kalpravidh, W., Gopinath, C.Y., Morzaria, S., 2014. Development and Application of a Vaccination Planning Tool for Avian Influenza. Avian Dis. 58, 437–452. doi:10.1637/10827-032414-Reg.1

5) The authors produce predictions that are many miles away from any previously reported occurrences – whilst the dotted lines on Figure 4 help the reader be aware of this, the authors should also consider a multivariate environmental similarity surface approach to gauge the extent to which these records are extrapolation, and where predicted values fall within environmental space already considered by the presence points already in place. There is an example of its usage in Elith et al. 2010 http://onlinelibrary.wiley.com/doi/10.1111/j.2041-210X.2010.00036.x/full

Thank you for this suggestion. We estimated the MESS for our different set of predictors and for our two set of occurrence points, and those are now added as supplementary information. As can be seen from the plot, we had very limited geographical areas with high predicted suitability outside of the range of values found in the occurrence points, and the range in the full set of points (positive and pseudo-absence) would only be larger.

*6) We are concerned about the independence of the occurrence points, particularly with respect to secondary spread rather than an inherent feature of the underlying environment in these locations. An example of where this is problematic would be charting the progress of Nipah outbreaks in Malaysia in 1998 – pigs and humans over a widespread area were infected yet in terms of environmental suitability, only one location, where the index case arose, is suitable for inclusion. All other cases are more appropriately modelled by understanding the nature of swine connectivity via livestock trade and human-animal interactions. To what extent are these influenza cases located, and to what extent are they environmentally driven?*

We believe that there was an element of misunderstanding here. Our suitability maps depict suitability for infection, including secondary spread, and not HPAI H5N1 or H5Nx clade 2.3.4.4 introduction or emergence events alone. This is precisely why we included anthropogenic and poultry factors, i.e. these are more likely to explain patterns of spread than environment variables, as our results quantitatively demonstrate. We agree that there is spatial dependence in the occurrence points, but this is precisely why we implemented the spatial cross- validation, so that we would not overestimate the goodness of fit of our models through correlated validation sets.

*7) This exposes a broader issue relating to the inclusion of the poultry layers in the model. Niche maps are simply reflections on the potential for presence of the pathogen – by using data from poultry and then including poultry measures as predictors it is therefore unsurprising that they have a high explanatory power within the model, particularly where there may be ineffective control for the biasing effects this may have. As a consequence, the maps may be a more accurate prediction of where outbreaks are more likely to occur (or be reported) rather than an accurate depiction of potential occurrence (i.e. anything from singular cases to very many). It was interesting therefore to see how the inclusion/exclusion of different covariates impacted this – even exploring what impact that has when converted into a map would be interesting to see, particularly an output based just upon Sets 2 and 3 alone.*

We agree, but a particular aspect of HPAI H5N1 is that the probability of presence was never found to necessarily scale-up with chicken numbers (where the disease is visible because of high mortality), but rather scales up with duck density (where the disease circulates more silently). This was very apparent in the review by Gilbert & Pfeiffer (2012, doi 10.1016/j.sste.2012.01.002). In many separate countries and studies, the probability of presence of HPAI H5N1 at different levels was not associated with high chicken density, most likely because of many different production systems associating with different patterns of transmission. In fact, an important point of the paper is precisely that for an infectious disease like HPAI H5N1, host-related variables are much better at spatially predicting disease occurrence than eco-climatic ones. This was evidenced by the fact that models combining host variables with other environmental predictors did not produce significantly better results when evaluated through spatial CV.

We agree that the outputs of Set 2 and 3 alone would be of potential interest to the readers and have included them as supplementary information.

[Editors' note: further revisions were requested prior to acceptance, as described below.]

[…]

Reviewer #2:

*The authors’ response to reviewer comments have clarified some initial concerns as well as provided interesting additional analyses. However, I do have some still outstanding issues, as well as some newly raised concerns given the authors response.*

It was interesting to note the use of gazetteers to approximate disease occurrence since this adds an additional concern as to how reliable this classification is, and if in doubt, what steps were taken to mitigate (e.g. turning specific lat longs into larger polygons capturing such uncertainty).

This could have been a concern if the analysis was carried out with a finer grain, and in fact, we implemented bootstrapping to sample through polygons in a previous study carried out in the Mekong region only (Artois et al. 2016 doi 10.1038/srep30316), at a spatial resolution of 0.0083333 decimal degree (approx. 1 km at the equator). In this case, we are using 17,068 records spread over three continents, at a spatial resolution of 0.083333 decimal degrees (approx. 10 km), so the influence of the spatial uncertainty of a fraction of these points should be quite negligible. For example, only 3.5% of the records of Empres-I were reported at the coarsest administrative level 1, 5.8% at administrative level 2, and the rest of the records were either at administrative level 3, or from locations of unknown quality for which little can be said about the position uncertainty. We’ve looked at this more carefully for national data sets that we know better, and Thailand alone recorded over 1500 records, all linked to administrative unit level 3, which have a median size of 16 km^2^, i.e. far smaller than our pixels. The scaling was somewhat similar for Vietnam, with outbreaks geo-referenced at the commune level. The addition of a bootstrapping procedure accounting for spatial uncertainty would be technically feasible, but would only apply for the records with a known quality, and we feel that given the strong spatial autocorrelation between neighboring pixels in most of the predictor variables, it would add a significant burden of processing for a negligible difference.

*The authors state that different pseudo-absence models produced "similar results", but provide no means for reviewers to gauge that similarity.*

We simply did not want to flood the reader with supplementary information material. A long and complex modeling procedure as this one entails many choices made at different points. Showing, all the possible consequences of the choices become intractable and of somewhat limited use. The figures below show the predicted distribution of the pseudo-absences distributed according to human populations (Figure 7) and poultry (Figure 8), with the IsChina term set to 0.

Author response image 1.**DOI:**
http://dx.doi.org/10.7554/eLife.19571.018

Author response image 2.**DOI:**
http://dx.doi.org/10.7554/eLife.19571.019

The differences are hardly noticeable (see, for example the small differences along the Iraq/Iran border) and we would be happy to add these predictions to the manuscript should the editor feel this is a necessity, alongside other results, but we feel this would simply add an extra load of results with limited added informational value.

*The clarification the authors present that these outputs are depicting suitability for infection, including secondary spread, to my mind is problematic, since this requires mixing different types of data together within the same model. It is therefore unclear to me how the disparate effects of environment (likely influencing primary infection) and human/movement/transmission networks (likely influencing secondary infection) will be disaggregated. Whilst the cross-validation may mitigate the potential clustering that primary and secondary cases have, I'm still not 100% convinced that this differing process can be adequately captured simultaneously. To my mind a more powerful demonstration of difference (or indeed lack of) would be a comparison of primary and secondary data alone.*

The distinction between primary and secondary infections is simply not straightforward for a HPAI virus. For example, HPAI H5N1 is depicted to have emerged in China in the late nineties, and only in the years 2003/2004 it started to spread internationally across Southeast Asia. What is the primary event in this situation? The emergence point in southern China? Or the first time it was introduced in these surrounding countries? If the former is considered, there’s only one event, i.e. the emergence point of the new HPAI H5N1 virus. If the latter is considered, these primary introductions in Vietnam, Thailand or even Indonesia may have as much to do with human and poultry trade network as they had with the environment. Or maybe the reviewer refereed to the first introduction of an HPAI H5N1 virus in a new area through wild birds? But how could one make such distinction out of pattern analysis?

A distinction between primary and secondary cases for HPAI would have to be based either be done using space-time filters to detect the first record in a particular area, which could introduce some bias by how it is defined (which area? how big? a continent? a country? a province? which time windows?), or would need to be based on some molecular definitions allowing to trace the hosts through which the virus transmitted, which would require individual sequences and extensive phylogeographic processing.

And even if this distinction was feasible, it wouldn’t necessarily allow differentiating those processes. Human movements and poultry trade transmission networks can also be at the origin of novel introductions in a country, and have been in several cases, and, conversely, short-distance contagious spread could also involve resident birds or environmental factors (e.g. water-borne, wind-borne).

This is probably why, out of the 47 papers on risk factor modeling of HPAI H5N1 reviewed in Gilbert & Pfeiffer (2012, doi: 10.1016/j.sste.2012.01.002), only one tried to make such distinction based on a space-time filter windows in Thailand. Studies that have looked at HPAI H5N1 outbreaks by type (in poultry vs. wild birds) are more frequent, but this doesn’t apply here, as our study only included records of HPAI found in poultry.

*It was good to see the inclusion of Set 2 and Set 3 results – what was disappointing however was that there was no discussion of the differences this produces (sometimes considerable e.g. India for HPAI). How much of this variation is due to differences in ratios of primary v. secondary cases considered as occurrence records across the different geographies for instance? I found it very interesting to see such a different picture result when the human/host variables were dropped.*

The paper is already fairly long and contains several messages, so we didn’t want to dwell too much into discussing the full set of results of models that we demonstrate to be poorer.

We don’t think it has to do with the ratio of primary/secondary cases. Rather, the most likely cause of the difference is the near absence of domestic ducks in the country, except in the few states surrounding Bangladesh. As discussed, domestic ducks play an important role in the epidemiology of HPAI H5N1 as silent carriers. So, when the poultry variables are dropped, the land-use model cannot make the difference between cropping/agricultural areas where ducks are abundant from those where they are scarce, and it therefore predicts higher risk in India than the host-based model, with poorer general performances. The resulting risk map is in fact in fairly good accordance with an analysis published previously and looking specifically at the situation in South Asia (Gilbert et al. 2010, doi 10.1007/s10393-010-0672-8 & Dhingra et al. 2014 doi 10.1016/j.sste.2014.06.003). See, in particular, Figure 1B in Gilbert et al. (2010) with the distribution of ducks in South Asia.

We have added a couple of lines to explain this in the Discussion (fifth paragraph) as we agree it is an illustrative example to explain why the host-based model could provide better extrapolations than the land-use one.